# An Analysis of the Probabilistic Track of the IPC 2018

**Florian Geißer**
Australian National University, Australia
florian.geisser@anu.edu.au

**David Speck**
University of Freiburg, Germany
speckd@informatik.uni-freiburg.de

**Thomas Keller**
University of Basel, Switzerland
tho.keller@unibas.ch

## Abstract

The International Planning Competition 2018 consisted of several tracks on classical, temporal and probabilistic planning. In this paper, we focus on the discrete MDP track of the probabilistic portion of the competition.

We discuss the changes to the input language RDDL, which give rise to new challenges for planning systems, and analyze each of the eight competition domains separately and highlight unique properties. We demonstrate flaws of the used evaluation criterion, the IPC score, and discuss the need for optimal upper bounds. An evaluation of the three top-performers, including their post-competition versions, and a brief analysis of their performance highlights the strengths and weaknesses of the individual approaches.

## Introduction

Since 1998, the International Planning Competition (IPC) empirically evaluates state-of-the-art planning systems on various benchmark problems to promote research and highlight key challenges in AI planning. Initially, the competition focused on classical planning, but other, less restrictive competition tracks have emerged in subsequent years. Various tracks for reasoning under uncertainty have been added in 2004, and the probabilistic planning track, which is the focus of this paper, was organized for the sixth time in 2018.

Many different planning techniques have been applied by the participants over the years, ranging from determinisation-based re-planning (Yoon, Fern, and Givan 2007) over reinforcement learning with policy gradient methods (Buffet and Aberdeen 2007) and incremental policy refinement (Teichteil-Königsbuch, Kuter, and Infantes 2010) to Monte-Carlo tree search (Keller and Eyerich 2012; Keller and Helmert 2013). Not only planning techniques, but also benchmark problems have evolved over time: e.g., the dominance of determinisation-based approaches was answered with a shift to probabilistically interesting problems (Little and Thiébaux 2007), or the probabilistic PDDL dialect PPDDL (Younes and Littman 2004) was replaced by RDDL (Sanner 2010).

The latest probabilistic track also introduced a few changes. Providing challenging problems not only for the competition but also for future years was one of the announced aims of the organizers, so twice as many instances were created for each domain, scaling up to larger state and action spaces, and the used subset of RDDL was extended to support action preconditions, finite-domain, and intermediate variables to be able to model more realistic planning tasks. Additionally, time and memory limits were increased to allow offline approaches that compute an executable policy to compete with the predominant online approaches that interleave planning for a single state and execution of the last decision.

Five different planning systems participated in the competition. As up to two versions of each planning system could be submitted, seven planner variants were evaluated in terms of the IPC score in conjunction with the two PROST (Keller and Eyerich 2012) versions that won IPC 2011 and 2014 as a baseline. Although each participant had a unique approach, no planner dominated in every domain and the final results are very close – the winner, PROST-DD (Geißer and Speck 2018), and the two runner-ups, SOGBOFA (Cui and Khardon 2018) and Random Bandit (Fern, Issakkimuthu, and Tadepalli 2018) are separated by roughly five points.

In the first part of this paper, we describe the changes of IPC 2018 and analyse whether the competition lives up to its claim and introduces benchmarks with the potential of providing challenges for MDP research in upcoming years. We analyse the competition domains and provide properties that set them apart from other IPC domains. The second part of our work focuses on the evaluation metric that was used to determine the winner of the competition. We demonstrate that the IPC score is flawed if optimal state values of a problem are unknown, and we hope to spark a discussion among the planning community on this topic. The final part of the paper analyses the performance of the planners on the IPC 2018 benchmarks. We compare the results of bugfixed versions of the competition's top-performers and give insight into their strengths and weaknesses.

## Input Language of the IPC 2018

As in the competitions in 2011 and 2014, the domains and instances of IPC 2018 were modeled in RDDL (Sanner 2010). In RDDL, both states and actions are described compactly by disjunct sets of parameterized variables and MDPs are specified as a dynamic Bayesian net with intermediate layers. There were some minor tweaks to the input language that are of no interest to this paper. The major changes on action preconditions and the introduction of finite-domain and

intermediate variables have an impact on the competition results and are discussed in the following.

**Action preconditions.** All RDDL domains from previous competitions come with a `state-action-constraints` section that contains a finite set of formulas $P$ in first-order logic over the set of state- and action variables. Formulas containing at least one action variable form a constraint on the set of applicable actions $a$ in a state $s$: if $s, a \not\models p$ for at least one $p \in P$, $a$ is not applicable in $s$. Formulas without action variables, on the other hand, were introduced to provide invariants to a planning system and could be ignored by a planner. As the semantics of state-action constraints were never formally specified, it was unclear if an action is inapplicable in a state if its application can lead to a state where an invariant is violated. This cannot be checked efficiently by a planner as the number of outcomes can be prohibitively large. In previous competitions, this was irrelevant as all state-action constraints were static. For IPC 2018, state-action constraints were hence replaced by an `action-preconditions` section that contains formulas that have to be considered by each planner and checked on the current state before an action is applied, and an `invariants` section that may be ignored.

In previous competitions, the number $\overline{A} \in \mathbb{N}$ provided in the `max-nondef-actions` specification of RDDL instances has also been used to describe the applicability of actions. As it is possible to translate this for a given set of action variables $\{a_1, \ldots, a_n\}$ to the action precondition $\sum_{i=1}^{n} a_i \leq \overline{A}$, the `max-nondef-actions` entry was removed for simplicity of notation. A side effect of dynamic action preconditions is that action variables carry significantly more parameters at IPC 2018 than in previous competitions to be able to "connect" action preconditions and transitions functions (conditional probability functions or CPFs in RDDL). In turn, this leads to a higher average number of ground action variables in the IPC 2018 instances and, due to the exponential relationship between action variables and actions, to a significantly higher number of actions if action grounding is performed naively. While the set of actions that is applicable in at least one state (i.e., the actions that need to be grounded) is significantly smaller in most instances, it is in general not tractable to compute this set exactly, as determining if a given action is legal in at least one reachable state is at least as hard as the bounded plan existence problem in classical planning. Previously, a small value for $\overline{A}$ has helped to keep the number of ground actions small even if the grounding procedure is simple. This safety net has been removed along with the `max-nondef-actions` section, and new techniques need to be developed for the challenge of grounding RDDL actions. To make grounding simpler, the IPC 2018 domains guarantee that an action $a$ with true action variables $A$ is inapplicable in all reachable states if there is another action $a'$ with true action variables $A' \subseteq A$ that is inapplicable in all reachable states.

**Intermediate variables.** A RDDL concept that has not been considered at IPC 2011 and 2014 are intermediate variables, which are typically used to determine the outcome of multiple interdependent stochastic effects. To illustrate the concept, assume that there is a 50% chance that two variables $v_1$ and $v_2$ are both true in the next state, and both are false otherwise. Modelling this with CPFs $v_i' = $ Bernoulli(0.5) for $i \in \{1, 2\}$, where Bernoulli is a RDDL keyword representing a Bernoulli distribution, would result in a model where all four possible value combinations come up with a probability of 25%. This does not reflect the desired transition dynamics. An intermediate variable $v$ with CPF $v = $ Bernoulli(0.5) and CPFs for $v_1$ and $v_2$ of the form $v_1' = v$ and $v_2' = v$ lead to the described model, though.

Some of the IPC 2018 domains are modelled with intermediate variables, but planners were allowed to choose between a domain version with or a compilation without this feature. The compilation is performed by replacing intermediate variables with state variables and adding artificial intermediate decision steps where only a dummy action `proceed-interm-level` can be applied. There are further details to this compilation, e.g., variables are introduced to remember which action was executed and to represent the current level, and the horizon is increased according to the levels of the intermediate variables. However, these are not relevant for this paper and hence omitted.

**Finite-domain variables.** Finite-domain variables can be modelled in RDDL as enum-valued variables, a feature that has not been used in previous competitions. IPC 2018 made this feature available to planners that support it, but also provided a compilation of finite-domain variables into binary variables. In the classical planning setting, such a compilation can be performed by replacing each finite-domain variable $v$ with domain $\text{dom}(v) = \{x_1, \ldots, x_n\}$ with binary variables $v\text{-is-}x_i$ for $i \in \{1, \ldots, n\}$. In the probabilistic setting, it is possible to do the same replacement, but the blowup is significantly larger. To illustrate this, consider a finite-domain variable $v$ of type `enum_type` which is defined over the values $x_1$, $x_2$ and $x_3$, and assume $v$ takes each value with uniform probability (modelled with the RDDL keyword `Discrete`). Due to the implicit dependency between the three values, we have to make sure that exactly one value becomes true in the next state, and a translation to

$$v\text{-is-}x_i' = \text{Bernoulli}(0.\overline{3});$$

for $i \in \{1, 2, 3\}$ is hence not correct. Instead, we have to sample these values consecutively, each time conditioned on the variables already sampled. As state variables are sampled at the same time, intermediate variables of the form

$$v\text{-is-}x_1 = \text{Bernoulli}(0.\overline{3});$$
$$v\text{-is-}x_2 = \neg v\text{-is-}x_1 \cdot \text{Bernoulli}(0.5);$$
$$v\text{-is-}x_3 = \neg v\text{-is-}x_1 \cdot \neg v\text{-is-}x_2;$$

are used in the compilation to model the consecutive sampling (in increasing index order) properly. In this form,

- $v\text{-is-}x_1$ becomes true with probability $\frac{1}{3}$
- the Bernoulli(0.5) case of $v\text{-is-}x_2$ becomes relevant in the $\frac{2}{3}$ of the cases when $v\text{-is-}x_1$ did not become true, resulting in a probability of $\frac{2}{3} \cdot \frac{1}{2} = \frac{1}{3}$ and

- $v$-is-$x_3$ becomes true if neither of the former became true and hence also with probability $\frac{1}{3}$.

If the term inside the Bernoulli statement cannot be simplified as much as here, these terms quickly grow very large.

## Competition Domains

In the following, we briefly introduce the domains that were used at IPC 2018 and highlight properties that make the domains particularly challenging. We base the presented information on the following sources[1]: 1) the output of a modified version of the PROST parser, which was enhanced with expert knowledge in some domains; 2) a random walk planner that computes averages over all states that are encountered in 200 runs; 3) and from computations by hand. In general, each domain consists of 20 instances, where the instances increase in size (in terms of states and actions), although not monotonically. While this has also been the case at IPC 2014, the largest instances of IPC 2018 (except for PUSH YOUR LUCK) are several orders of magnitude larger than the smallest ones.

**ACADEMIC ADVISING** is the only domain of IPC 2018 that has appeared in a previous competition, and it is equivalent to its predecessor apart from some minor changes that became necessary due to altered competition rules. However, neither of the 20 instances has been used before. In ACADEMIC ADVISING, a student takes courses at a given cost, aiming to complete a predefined subset of courses. The probability to pass a course increases with the number of previously passed prerequisites.

Prior to IPC 2018, ACADEMIC ADVISING has already been the domain with the largest number of applicable actions. However, if we compare the largest instances, that number grew from 466 in 2014 to more than $10^{11}$ in 2018, and the median over the instances increased from 43 to 1862, which poses a real challenge to the competitors.

**CHROMATIC DICE** is an MDP variant of the popular dice game Yahtzee, where up to five dice are rolled up to three times and show both values and colors (determined by independent stochastic processes). After rolling, the planner has to select a category and receives a reward depending on the faces of the dice and the selected category. At the end of the interaction, the planner may receive various bonuses if it performed well in certain category sections.

CHROMATIC DICE is special because it has by far the largest reward formula among all competition domains, consisting of formulas over almost 10000 state variables (most occur multiple times, but almost all state variables are among them at least once) and 24 (different) action variables. For a near-optimal policy, the boni become very important, and the planner has to plan ahead exceptionally far.

---

[1] Zenodo link for complete dataset used in this paper: http://doi.org/10.5281/zenodo.3235174

**COOPERATIVE RECON** is a significantly altered variant of the IPC 2011 domain RECON. In the 2018 version, the planner controls one or more planetary rovers that examine objects of interest in order to take a picture of detected life.

In most instances, there are several rovers, and collaboration between them is a key challenge. Rovers carry different equipment and have to share tasks among them to succeed, and they are also able to support other rovers in their tasks for a higher probability of success. This makes COOPERATIVE RECON special because the mutual application of two action fluents can be more valuable than the sum of its parts.

**EARTH OBSERVATION** is based on the domain by Hertle et al. (2014) where the planner controls a satellite orbiting Earth that has to take pictures of the landscape below, taking into account the current weather forecast (the presence of clouds when a picture is taken leads to poorer image quality and hence results in a lower reward).

While the branching factor induced by actions is very low in this domain – there are only 4 actions to move the camera and take a picture – the branching factor due to uncertainty is immense. The weather can change the current cloud cover drastically, and the number of successor states of a given state-action pair is tremendous, comparable only with the SYSADMIN domain of IPC 2011.

**MANUFACTURER** is a domain where the agent manages a manufacturing company that buys goods to use them in the production of other goods. The domain is modular in the sense that more and more options become available the more challenging the instance is. In the smallest instances, the agent only buys, produces and sells goods. More complex instances allow the construction of additional factories, hiring staff to influence the price or contracting a manager who enables the execution of more efficient actions.

All modules have in common that the agent has to accept an immediate cost for an increased long-term reward. This is already true for the basic buy-produce-sell cycle, and the horizon until the investment pays off gets larger and larger with more challenging instances. Additionally, this is one of the domains with the largest number of applicable actions and relevant preconditions (more than $10^6$ in both cases).

**PUSH YOUR LUCK** is a single-player game where the main challenge lies in determining the optimal moment to stop a repeated stochastic process. The player rolls one or more dice repeatedly until they select to cash-out, yielding a reward that corresponds to the product of all rolled values. However, if the player plays too risky and a number comes up a second time, all rolled values are reset.

The instances of PUSH YOUR LUCK are among the smallest of IPC 2018. However, better play is rewarded disproportionately due to the exponential growth of the reward in the case of success and the total reset in the fail case.

**RED-FINNED BLUE-EYE** are a species of fish that are endemic to seven artesian springs in the Edgbaston Reserve in

Central Queensland, Australia. The species is critically endangered due to competition by the invasive Gambusia. This domain is inspired from the work of Nicol et al. (2017). A planner has to make sure that red-finned blue-eye do not become extinct, either by removing or poisoning Gambusia or by translocating red-finned blue-eyes. The domain is probabilistically interesting as the springs get connected only in the rain season depending on the (probabilistically determined) amount of rain.

RED-FINNED BLUE-EYE is challenging because it has the largest median number of actions (2680) and action preconditions (almost 10000), and more than $10^6$ actions in the hardest instances. The planning horizon of up to 120 and a median of 90 is also the largest, which is particularly relevant because extinction of red-finned blue-eye leads to a disproportionately high penalty.

**WILDLIFE PRESERVE** is inspired from work of Nguyen et al. (2013) and Fang, Stone, and Tambe (2015) on rangers that protect a wildlife reserve from poachers by sending rangers to specific areas. Poachers attack parts of the reserve depending on their preferences and an expectation where rangers will likely show up. This expectation is computed by exploiting the assumption typically taken in Stackelberg Security Games that the defenders' (i.e., rangers) mixed strategy is fully observed by the attacker and memorized by poachers for a predefined number of steps.

In each step, the planner obtains a reward for each area that has not been attacked undefended, and a penalty for each area that has. The challenge is to predict where poachers will attack with high probability and to lure poachers into attacking an area where they are caught. Determinisation-based policies are informative in instances where the memory of poachers is short, but the quality decreases quickly when poachers remember more of the rangers' decisions.

## Participants

We briefly introduce the competition participants and the underlying techniques they use. More details can be found in the planner abstracts that can be found on the competition website[2].

**PROST** (Keller and Eyerich 2012) is the winner of the two previous IPCs in 2011 and 2014 and participated non-competitively to serve as a baseline. PROST 2011 and PROST 2014 differ mostly in the used search algorithm: the former is based on the popular UCT algorithm (Kocsis and Szepesvári 2006), while the latter applies the UCT$^\star$ algorithm of Keller and Helmert (2013). Both baselines use an iterative deepening search (IDS) on the (most-likely) determinised task to initialize action-values of search nodes that are added to the search tree.

The versions that were used for IPC 2018 are not exactly the same planners that competed in the previous competitions: bugfixes provided over the last few years were incorporated and a simple linear time distribution of the remain-

[2]ipc2018-probabilistic.bitbucket.io

ing time was used for each step. Additionally, the parser was updated to exploit the guarantee on action preconditions described before: the implementation starts to check potential applicability of an action with actions where only one action variable is true, and it iteratively adds more variables until a precondition is violated independently from the state.

**PROST-DD** is based on PROST 2014 and applies the UCT$^\star$ algorithm. The planner differs from the baseline in the underlying action-value initializer function, the recommendation function used to select the action applied in each step, and does not use the baseline parser implementation. Instead, the PROST-DD parser performs resolution- and backtracking-based search in a similar fashion to the DPLL algorithm (Davis, Logemann, and Loveland 1962). Additionally, the performance of the evaluation of action preconditions was improved. As recommendation function the planner uses the *most played arm* recommendation (Bubeck, Munos, and Stoltz 2009), which favors the action which was selected the most in the root node of the search (the baseline planners favor the action with the best expected outcome).

For the heuristic function, the planner symbolically represents a deterministic version of the planning task as Algebraic Decision Diagrams (Bahar et al. 1993). It computes step-wise estimates in a similar fashion to symbolic value iteration (Hoey et al. 1999) and symbolic backward search (Speck, Geißer, and Mattmüller 2018). If the symbolic computation is not able to compute the estimates for a number of steps that is equal to the problem horizon, the planner performs iterative deepening search instead. Two versions of PROST-DD participated in the competition, which differ in the determinisation that is used in the heuristic: outcomes with probability smaller than $0.5$ are pruned in one version, and smaller than $0.1$ in the other.

**Random Bandit** is built upon the PROST framework and is based on the $\epsilon$-greedy algorithm for multi-armed bandit problems, which estimates state values by simulating the greedy action with probability $1 - \epsilon$ and a random action otherwise. The parameter $\epsilon$ is set to $0.5$. This decision in the root node is followed by a random walk whose simulation depth is initially determined as the minimum of 7 and an estimate that is based on the time required for IDS on the most-likely determinised task.

**Conformant SOGBOFA** is based on the work by Cui and Khardon (2016) and Cui, Marinescu, and Khardon (2018). It symbolically represents the state value function of the current state as an abstract syntax tree and searches for the best action by calculating gradients on this symbolic representation by means of automatic differentiation. One property of this representation is that computations are performed on the lifted action representation, which allows the planner to deal with large action spaces. Therefore, the planner does not explicitly ground actions. Two different versions of the SOGBOFA system participated in the competition: SOGBOFA-F and SOGBOFA-B. They differ in the way fractional values in the rollout policy are treated.

**A2C-Plan**   While all previous planning systems perform planning online, A2C-Plan is an offline planner working in two phases: in the training phase, a deep neural network is trained, using an actor-critic algorithm (Konda and Tsitsiklis 1999) which combines learning of a policy network and updating its parameters via gradient updates. The planner is built upon the PROST framework and therefore uses the baseline parser implementation.

**Imitation-Net**   is another offline planner based on neural networks. It follows a supervised learning approach which generates training data by following a greedy one-step policy based on random sampling. Based on this training data, a policy network (Issakkimuthu, Fern, and Tadepalli 2018) is trained using stochastic gradient descent, which is used to select the best action in each step. Similar to A2C-Plan, the planner is built upon the PROST framework.

## On Evaluation Metrics

The evaluation of the planner performance was in principle the same as in the previous competitions: planners performed a sequence of interactions (runs) with rddlsim (Sanner 2010) that simulate the execution of the planner's policy, and the average cumulative reward over those runs is used to estimate planner performance. However, some details were changed in comparison to previous competitions: the number of runs was increased from 30 to 75 for higher statistical significance of the averages; the number of instances per domain was doubled from 10 to 20 for better scaling between small and large instances; the horizon was instance dependent to add an additional challenge with short or large horizons; memory was doubled from 2GB to 4GB to reflect modern hardware; and the average deliberation time per step was raised from approximately 1 second to 2.5 seconds, resulting in a total deliberation time between approx. 1 hour for the instances with the smallest horizon of 20 and more than 6 hours for the instances with the largest horizon of 120, an amount of time that allows offline planners to come up with a policy that is competitive with online planning. Furthermore, planners had to announce prior to each run if the round contributes to the evaluation to ensure that an optimal policy also maximizes the average cumulative reward over the executions of the policy (Keller and Geißer 2015).

**IPC Score.**   The quality of each planner is measured in terms of the IPC score with an artificial minimum policy that is set to the maximum of a random policy and (if possible) a policy that only executes no-op actions. The IPC score is a normalized value between 0 and 1, where 0 is assigned to a planner that did not simulate 75 runs successfully or did not outperform the minimum policy, 1 is assigned to the best planner in the competition (if there is a planner with 75 valid runs that performed better than the min policy) and a value that is linearly interpolated between these extremes is assigned to each of the other planners. In the following, we argue that without having access to an optimal upper bound, i.e., the average cumulative reward of an optimal policy, a planner evaluation based on the IPC score is flawed. We

| Planner | A | B | Min. | Opt. |
|---|---|---|---|---|
| Instance 1 | 5.00 | 1.00 | 0 | 100 |
| Instance 2 | 1.00 | 2.00 | 0 | 2 |
| IPC Score w/o Opt. | **1.50** | 1.20 | 0 | - |
| IPC Score w. Opt. | 0.55 | **1.01** | 0 | 3 |

Table 1: A hypothetical competition result, where the IPC score is evaluated with and without consideration of optimal upper bounds.

want to emphasize that the aim of this discourse is not to question which planner is the winner of the competition, but to spark a discussion among the community on the underlying evaluation metric, and to motivate further research on upper bounds for the benchmark sets used in planning.

In principle, the problem is that without an optimal upper bound the IPC score is not a stable metric, as the introduction of a new participant may change the results of the scores of all other participants if the new participant performs best in *any* instance. A consequence is that introducing a new participant may change the ranking between the current participants. Even worse: after introducing optimal upper bounds on the rewards the winner might change.

Table 1 shows an example where the normalization with optimal rewards changes the outcome of a hypothetical competition. Here, without considering optimal expected reward, the performance of planner A appears to be better than the performance of planner B. More precisely, A is five times better than B in instance 1, while B is only twice as good as A in instance 2. According to the IPC score, planner A is the winner of the competition. After introducing upper bounds on the average cumulative reward, it turns out that planner B performs optimally in the second instance and in the first instance both planners perform poorly in comparison to the optimal policy. Now, according to the IPC score, planner B is the winner of the competition. Note that the order can also change with the introduction of a new planner who performs better in instance 1 than planner A: submitting similar planner configurations to the competition might be detrimental for the planner performance. With optimal upper bounds, the order of two planners can never change when a new planner is introduced, since the IPC values of these two planners do not change.

In the absence of optimal expected reward values, one option is to rank the planners in order of performance on each instance, instead of accumulating the relative average rewards. Such a ranking system is closely related to the field of social choice theory. Each instance induces a ranking which can be interpreted as an independent preference relation. In other words, each instance represents a voter and each planner a possible candidate. There are several election methods for determining a winner in such a ranking system. Each fulfill different criteria, such as independence from irrelevant alternatives, independence of clone alternatives, or the majority criterion. Yet, it is a well-known game theoretical result that no voting system can fulfill even a small number

of particularly important criteria (Osborne and Rubinstein 1994). Therefore, before any voting system can be applied, it is important to define the criteria one wants to meet.

While such a ranking system ignores the relative performance of the planners, the discussion shows that even for arguably simpler measures the final evaluation depends on the underlying evaluation criteria one wants to meet and guaranteeing multiple criteria might even be impossible. We also did not address the probabilistic aspect of the problem, where we have to consider variance on the possible rewards. Designing a satisfactory system is not only out of the scope of this paper, but also requires the input of the IPC community and a theoretical analysis and discussion of this topic.

## Analysis of Planner Performance

One should keep in mind that the aim of the planning competition is not necessarily to elect the best planning system, but to promote research and highlight current challenges in planning. Thus, we now take a closer look at the competition results and discuss the strengths and weaknesses of the different systems. While the IPC score might not be the best metric to determine the best planning system, it still gives some insight into the planner performance on individual domains. In the following we will put our focus on the online planning systems, as both offline planners performed worse than the online planners in almost[3] all domains (total IPC score of 28.6 for Imitation Net and 26.6 for A2C-Plan), and leave a study of these systems for future work.

We begin our analysis by having a look at the official competition results, depicted in Table 2. Since the baseline planners PROST 2011 and 2014 were only provided to compare the current participants with the winners of the previous competitions, the official winner was PROST-DD. The differences between PROST-DD, SOGBOFA and Random Bandit are minor, though. Furthermore, all submitted planners contained more or less severe bugs at the time of the competition, which affected their performance on some of the domains. Since the goal of our analysis is to shed light on the current state of the art in probabilistic planning and exhibit potential future work, we performed an additional evaluation within the same competition setup, but used an updated version of the planners if available. IPC scores with an updated version of PROST-DD and SOGBOFA (Cui, Keller, and Khardon 2019) are depicted in Table 3. Note that the scores are computed without considering A2C-Plan and Imitation-Net results.

It is apparent that the bugfixes for both planners were quite significant on some of the domains and both PROST-DD and SOGBOFA significantly outperform the baseline planners in the bugfixed versions. Furthermore, PROST, PROST-DD and SOGBOFA each dominate all other planners in at least one domain with respect to the IPC score. As a consequence, we further focus on a brief analysis of only these three planners, look into possible reasons for their performance, and highlight possible future research to provide additional insight.

---

[3]The only domain where Imitation-Net showed competitive performance was WILDLIFE PRESERVE.

**SOGBOFA**  is not only the planner which differs architecturally and algorithmitically the most from the otherwise PROST-based planners, the most recent version also significantly outperforms its competitors. The most prominent feature of SOGBOFA is the support of large action spaces (no grounding process is involved), thus one reason for the strong performance might be that the planner is simply able to at least do *something* on the larger domains. To see this, we compare the number of instances where each planner completed all 75 runs, depicted in Table 4. Indeed, there are 17 instances where SOGBOFA was the only planner able to complete all 75 runs. Note that none of the other planners uniquely completed 75 runs on any instance (considering only a single configuration per planner). However, recall that the IPC score is only affected if the planner outperforms the min-policy. This was only the case in 5 out of the 17 instances (1 COOPERATIVE RECON, 3 RED-FINNED BLUE-EYE, 1 MANUFACTURER), which indicates that the planner did not only perform well because it was able to handle many instances where the other planners failed.

The advantage of SOGBOFA in instances with large action spaces is further highlighted by its performance in RED-FINNED BLUE-EYE and the second half of the COOPERATIVE RECON instances. Both domains have a large action space, and although the other planners are able to outperform the min-policy in these domains, SOGBOFA has a significantly higher average reward. The third domain where SOGBOFA shines is the MANUFACTURER domain, yet the applicable action space of this domain is quite low (only a couple of actions are applicable in each step). As this is a domain where the SOGBOFA algorithm outperforms the THTS-based algorithms independently from the size of the problem, we see this domain as a candidate for future research on both algorithms. We also conjecture that for these three domains the relative performance indicated by the IPC score would still hold, even if an upper bound on these problems would be provided.

The only domain where SOGBOFA performed significantly worse than every other competitor is PUSH YOUR LUCK. Interestingly, this is the domain with the overall smallest problem size, both in terms of applicable actions and state-space size. This allows us to compute the optimal reward for some of the instances and compare it to the planner results, depicted in Figure 1. While THTS-based planners often reach the optimal expected reward, this does not hold for SOGBOFA, which might be attributed to the optimality guarantees of UCT*, whereas the automatic differentiation algorithm does not provide such guarantees.

**PROST-DD**  showed the best performance in ACADEMIC ADVISING and WILDLIFE PRESERVE. In the following we will focus on a comparison between the PROST-DD planner and the baseline, as both planners share the underlying algorithm but differ in initialization, recommendation function, and the grounding process. We will focus on the initialization and the grounding process, and refer to Keller (2015) for a comparison of the behaviour of the different recommendation functions.

|  | **PROST** | | **PROST-DD** | | **Random** | **SOGBOFA** |
|---|---|---|---|---|---|---|
|  | **2011** | **2014** | **0.1** | **0.5** | **Bandit** |  |
| ACADEMIC ADVISING (20) | 3.2 | 3.3 | 5.8 | **6.6** | 0.7 | 4.1 |
| CHROMATIC DICE (20) | 12.8 | 10.1 | 7.6 | 7.5 | 17.1 | **19.4** |
| COOPERATIVE RECON (20) | 9.0 | 10.7 | 10.3 | **12.0** | 1.5 | 6.9 |
| EARTH OBSERVATION (20) | 18.7 | **19.9** | 6.5 | 5.3 | 12.8 | 7.4 |
| MANUFACTURER (20) | **7.1** | 2.7 | 3.3 | 2.8 | 4.1 | 0 |
| PUSH YOUR LUCK (20) | 6.3 | 14.2 | **15.0** | 12.7 | 13.1 | 1.4 |
| RED-FINNED BLUE-EYE (20) | 6.9 | 6.0 | 5.9 | 5.4 | 5.6 | **18.3** |
| WILDLIFE PRESERVE (20) | 3.9 | 7.9 | **14.3** | **14.3** | 10.8 | 4.8 |
| **Sum (160)** | 67.9 | 74.7 | **68.8** | 66.5 | 65.6 | 62.3 |

Table 2: Official IPC scores of the top performers of the International Planning Competition 2018.

|  | **PROST** | | **PROST-DD** | | **Random** | **SOGBOFA** |
|---|---|---|---|---|---|---|
|  | **2011** | **2014** | **0.1** | **0.5** | **Bandit** |  |
| ACADEMIC ADVISING (20) | 4.06 | 3.33 | **6.84** | 5.61 | 0.81 | 4.86 |
| CHROMATIC DICE (20) | 12.91 | 10.04 | 9.82 | 10.49 | 16.97 | **19.2** |
| COOPERATIVE RECON (20) | 6.19 | 6.85 | 7.82 | 8.06 | 1.33 | **15.11** |
| EARTH OBSERVATION (20) | 18.76 | **19.73** | 17.24 | 16.77 | 12.74 | 11.52 |
| MANUFACTURER (20) | 4.05 | 2.08 | 3.08 | 4.72 | 1.14 | **10.34** |
| PUSH YOUR LUCK (20) | 6.57 | 14.61 | 14.99 | **15.22** | 13.22 | 2.32 |
| RED-FINNED BLUE-EYE (20) | 6.41 | 7.32 | 5.9 | 4.92 | 5.49 | **18.97** |
| WILDLIFE PRESERVE (20) | 3.99 | 7.98 | **15.87** | 14.99 | 10.78 | 8.68 |
| **Sum (160)** | 62.94 | 71.94 | 81.56 | 80.87 | 62.48 | **91.0** |

Table 3: IPC scores of the IPC 2018 top performers, based on updated planner versions.

|  | **PROST** | | **PROST-DD** | **Random** | **SOGBOFA** |
|---|---|---|---|---|---|
|  | **2011** | **2014** | **0.1** | **Bandit** |  |
| ACADEMIC ADVISING (20) | 11 | 12 | 14 | 11 | **20** |
| CHROMATIC DICE (20) | **20** | **20** | **20** | **20** | **20** |
| COOPERATIVE RECON (20) | 13 | 15 | 17 | 17 | **20** |
| EARTH OBSERVATION (20) | **20** | **20** | **20** | **20** | **20** |
| MANUFACTURER (20) | 10 | 11 | 11 | 11 | 16 |
| PUSH YOUR LUCK (20) | 9 | 17 | **20** | **20** | **20** |
| RED-FINNED BLUE-EYE (20) | 11 | 15 | 15 | 17 | **20** |
| WILDLIFE PRESERVE (20) | 4 | 9 | 16 | 16 | 10 |
| **Sum (160)** | 98 | 119 | 133 | 132 | **146** |

Table 4: Number of instances for each planner where all 75 runs were completed in time.

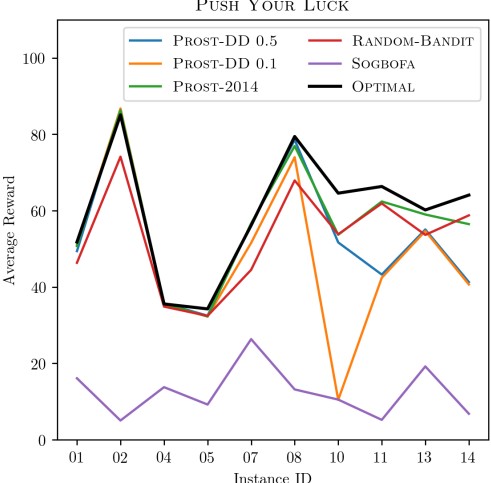

Figure 1: Average Reward in PUSH YOUR LUCK.

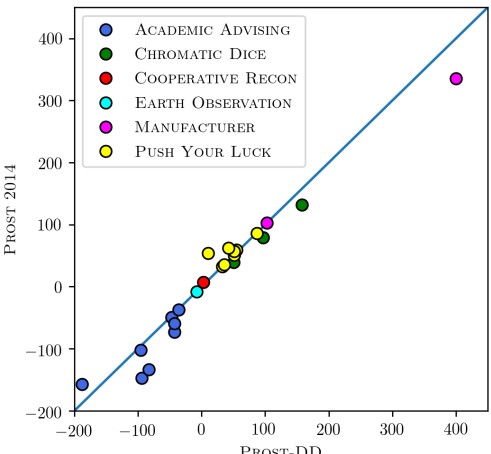

Figure 2: Average reward on instances where the DD heuristic was fully constructed.

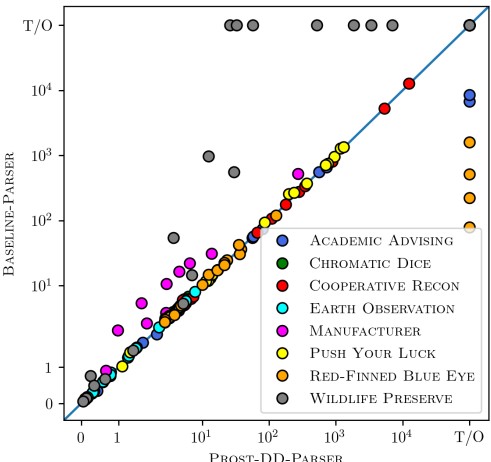

Figure 3: Parsing times in seconds.

As a first step, we evaluate the impact of the decision diagram based initialization. Note that in instances where the decision diagrams are not constructed, both planners rely on the same initialization (IDS). Figure 2 plots the average reward of PROST-DD and of PROST 2014 for each instance where both planners finished all 75 runs and where the decision diagram data structure was completely built up[4]. In general, the heuristic improves the average reward if constructed, which is also the reason for the stronger performance in ACADEMIC ADVISING. The varying performance in PUSH YOUR LUCK is a result of both heuristics being very uninformative: in instances where a dice has more than 6 faces the probability for each face to appear becomes 0 in the determinisation. It is worth to note that for instances where the heuristic was not constructed, PROST-DD wastes up to 12.5% of the search time.

Next, we analyse the impact of the parser difference. Both parsers generate the same grounded instance, but differ in execution time. Figure 3 depicts the parsing time in seconds per instance. Clearly, the advantage of PROST-DD in WILDLIFE PRESERVE is due to the timeout of the baseline parser in half of the instances. On the other hand, in some instances of RED-FINNED BLUE-EYE and ACADEMIC ADVISING the PROST-DD parser times out while the baseline parser is able to parse the instance. Due to the poor performance of the baseline on these instances this does not influence the final score, though.

**PROST.** We conclude our brief planner analysis with a few words on the performance of the UCT$^\star$ search algorithm, which is the core of both PROST 2014 and PROST-DD. The two domains where this approach significantly outperformed SOGBOFA are EARTH OBSERVATION and PUSH YOUR LUCK. For PUSH YOUR LUCK, we have already seen that part of the reason is the optimality guarantee given by the Bellman backups of UCT$^\star$, which allows to compute the optimal expected reward for many of the instances (this also holds for WILDLIFE PRESERVE). It would certainly be interesting to see how close the EARTH OBSERVATION results are to the optimal values. Additionally, EARTH OBSERVATION is the domain with the least number of actions: only 4 actions are applicable in every state, which apparently favors sampling-based techniques.

## Conclusion

To keep the conclusion brief, we emphasize the importance of having access to (near-) optimal rewards for the computation of IPC scores. Our analysis indicates that the benchmark set of IPC 2018 provides a challenge for the current state of the art in probabilistic planning and gives insight on possible future research. Future work on the PROST planner should be concerned with how to efficiently deal with large state and action spaces. For the SOGBOFA system, an interesting question is if it is possible to provide optimality guarantees. Why offline planners were unable to meet the performance of online systems remains an open question.

---

[4]We removed data points for WILDLIFE PRESERVE, as both planners share the average reward in 8 out of 9 instances.

**Acknowledgments.** Florian Geißer was supported by ARC project DP180103446, "On-line planning for constrained autonomous agents in an uncertain world". David Speck was supported by the German National Science Foundation (DFG) as part of the project EPSDAC (MA 7790/1-1). Thomas Keller received funding for this work from the European Research Council (ERC) under the European Union's Horizon 2020 research and innovation programme (grant agreement no. 817639).

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
