# OpenReview forum: "An Analysis of the Probabilistic Track of the IPC 2018"
_icaps-conference.org/ICAPS/2019/Workshop/WIPC_

### Official Review · AnonReviewer3 · 2019-04-25
**In-depth analysis of the probabilistic track**

**Rating:** 9
**Confidence:** 4

**Review:**

This paper analyzes the probabilistic track of IPC18, including an analysis of the changes
made with respect to previous editions, as well as the domains, participants, and results
of this edition. This is a clear fit for the workshop.

The analysis is very in-depth, and quite a good summary of everything related to this
track of IPC'18. I have only a few comments of details that I wondered about while reading
the paper:

- The input language section assumes certain familiarity with RDDL. Perhaps this is ok
  (most people interested in this will know the details of RDDL anyway) but the paper
  could be more accessible for people approaching the probabilistic track for the first
  time. For example, the discussion of max-nondef-actions says that this has typically
  helped to keep the number of grounded actions small, but it is not clear what type of
  information is provided there and how this helps the grounding process. Similarly, the
  paper refers to "action variables" without explaining what they are or how they are used
  in detail. Perhaps, giving an example here would be useful. At first I thought of them
  as the parameters of PDDL actions until I realized it was a different thing.

 - I want to highlight the first sentence of your analysis section because I really liked
   it: The IPC is a fun competition, but one should not only highlight the results of the
   winner but do a more broad analysis!

 - Regarding the results of the offline planners: do you have more insights about why they
   performed poorly? Do they tipically fail to derive a valid policy within the time
   limit?  Or is the policy very inferior to the online planners?

 - It is somewhat surprising to me that most planners cannot terminate the run. Why is
   that the case? Could this be solved by just fine-tuning some parameters (as the number
   of simulations performed before taking a decision) or setting a time limit for
   preprocessing time?

 - Regarding the comparison of random bandit vs Prost. Since it is built on top of the
   Prost framework, is the main difference the use of UCT vs random bandit? Or are the
   simulations also different? Could it make sense to do an analysis similar to the one
   Prost vs Prost-DD with random bandit as well to see how similar it is?

Minor comments:
  - A RDDL -> An RDDL
  - protect a wildlife preserve -> shouldn't be "wildlife reserve"? Not in the domain name (which refers to preserving wildlife in a reserve, I think).
  - the the -> the
  - might might -> might

---

### Official Review · AnonReviewer2 · 2019-04-25
**The 2018 probabilistic track, one year on**

**Rating:** 10
**Confidence:** 5

**Review:**

This paper presents an overview of the 2018 probabilistic IPC, going over the results and presenting additional insights (and further results) that have arisen since then.

This paper is a great addition to the workshop, and I applaud the authors for taking the time to get real insights from the IPC -- we see fewer 'papers on the IPC' than we used to, so it's good to have one in hand, and the workshop is a good place for it.  I particularly appreciated the efforts the authors went to obtain additional pertinent results, including a greater allowance for bug fixes and other planner changes than can practically be made in the IPC setting.

A few minor notes:

p2 - ''connect'' needs double back-quotes before the word connect, rather than ''
p3 - boni -> bonuses

(bonus comes from the Latin bonum (a good thing) not the Latin bonus (a good man), so doesn't inherit its Latin plural (which is indeed boni); unless you really did mean that the planner receives several good men if it performs well :) )

p5 - noop -> no-op